# Stochastic Sample Approximations of (Local) Moduli of Continuity

## Abstract

Modulus of local continuity is used to evaluate the robustness of neural networks and fairness of their repeated uses in closed-loop models. Here, we revisit a connection between generalized derivatives and moduli of local continuity, and present a non-uniform stochastic sample approximation for moduli of local continuity. This is of importance in studying robustness of neural networks and fairness of their repeated uses.

## 1 Introduction

There is a substantial interest in scalable verification of properties of neural networks. On one hand, some qualitative properties of a neural network can be established by straightforward inspection of its compute graph. For example, a ReLU-based network Householder (1941) will not be differentiable at points given by the break-points of the ReLU activation function, but will be locally Lipschitz. On the other hand, establishing quantitative properties of the neural network, such as the precise modulus of local Lipschitz continuity and related metrics such as Consistent Robustness Analysis (CRA), is often non-trivial.

The modulus of local continuity (also known as the Lipschitz constant), a key quantitative property of a neural network, is important both for evaluating the robustness (Bastani et al., 2016; Huster et al., 2018; Yang et al., 2022; Zühlke & Kudenko, 2024) of neural networks and for evaluating properties (Mareček et al., 2023; Zhou et al., 2024; Nazarov et al., 2025) of their repeated uses in closed-loop models. Virmaux & Scaman (2018) showed that estimating the modulus of local Lipschitz continuity is NP-Hard even for a two-layer ReLU-based network. First methods including "AutoLip" and "SeqLip" of Virmaux & Scaman (2018) propagated operator-norm surrogates along the computation graph. Recently, much of the research has focussed on two classes of methods. "LipMIP" and its variants (Tjeng et al., 2019; Anderson et al., 2020; Zhang et al., 2022; Schwan et al., 2023, e.g.) utilized mixed-integer programming formulations of the neural network, either directly for ReLU-based networks, or using some piece-wise linear approximation (Gurobi Optimization, LLC). "LipSDP" and its variants (Fazlyab et al., 2022; Chen et al., 2020; Latorre et al., 2020; Chen et al., 2021; Dvijotham et al., 2020; Chen, 2022; Xue et al., 2022; Wang et al., 2024; Yang et al., 2024; Pauli et al., 2024; Xu & Sivaranjani, 2024; Lan et al., 2022; Syed & Hu, 2025, e.g.) utilize relaxations in the form of semidefinite programming (SDP). Substantial progress has been made recently in scaling these approaches. State-of the-art implementations (Wang et al., 2024; Yang et al., 2024; Lan et al., 2022) can scale to shallow network with each of the hidden layers having hundreds of neurons. Still, this is substantially less than the neural networks utilized in many practical applications.

Here, we revisit a connection between generalized derivatives and moduli of local continuity, and present a non-uniform stochastic sample approximation for these. At its simplest, the connection is well known (Virmaux & Scaman, 2018; Goodfellow, 2018; Weng et al., 2018a; Sridhar et al., 2022). Sampling values from the Clarke subdifferential at inputs sampled uniformly at random is a benchmark traditionally used for comparison of both LipSDP and LipMIP. For instance, (Wang et al., 2024) sampled 500,000 points uniformly at random from the input space and computed the maximum norm, lower bound. We propose algorithms for estimating the modulus of continuity using non-uniform sampling from the input space utilizing upper-confidence-bound (UCB) policies. The resulting estimator is consistent, asymptotically unbiased, and asymptotically optimal within a class of sampling policies.

## 2 RELATED WORK

In this section we recall the basic theoretical setup for estimating Lipschitz constants, and review various ways this estimation has been done in practice. We roughly follow the notation and definitions as in (Jordan & Dimakis, 2020). In particular, we assume $\|\cdot\|_\alpha$ and $\|\cdot\|_\beta$ are norms on $\mathbb{R}^n$ and $\mathbb{R}^m$ respectively.

**Definition 1.** *(Jordan & Dimakis, 2020, Definition 1) The **local** $(\alpha, \beta)$-**Lipschitz constant** of a function $f : \mathbb{R}^n \to \mathbb{R}^m$ over a set $X \subseteq \mathbb{R}^n$ is defined as the following quantity:*

$$L^{(\alpha,\beta)}(f, X) := \sup_{x,y \in X} \frac{\|f(y) - f(x)\|_\beta}{\|x - y\|_\alpha} \quad (x \neq y)$$

*Moreover, if $L^{(\alpha,\beta)}(f, X)$ is finite, we say that $f$ is $(\alpha, \beta)$-**Lipschitz** over $X$.*

The starting point for estimating Lipschitz constants in terms of Jacobians $J_f(x)$ is the following well-known fact for smooth functions defined on convex[1] domains:

**Lemma 2.** *If $f : \mathbb{R}^n \to \mathbb{R}^m$ is $C^1$ and $(\alpha, \beta)$-Lipschitz over an open convex set $X$, then:*

$$L^{(\alpha,\beta)}(f, X) = \sup_{x \in X} \|J_f(x)\|_{\alpha,\beta}$$

*where $\|\cdot\|_{\alpha,\beta}$ is the induced matrix norm on $\mathbb{R}^{m \times n}$*

In practice, many functions, including neural networks involving ReLU activations, are not smooth but are still Lipschitz. At the points where such functions are non-smooth, one does not have a well-defined Jacobian; instead, one must resort to one of several (possibly set-valued) *generalized derivatives*, for instance the *Clarke Jacobian* $J_f^c$ (cf. Definition 27). In this case, Lemma 2 can be generalized as follows:

**Theorem 3.** *(Jordan & Dimakis, 2020, Theorem 1) Suppose $f : \mathbb{R}^n \to \mathbb{R}^m$ is $(\alpha, \beta)$-lipschitz over an open convex set $X$. Then the following equality holds:*

$$L^{(\alpha,\beta)}(f, X) = \sup_{x \in X, G \in J_f^c} \|G^T\|_{\alpha,\beta}$$

The modulus of local continuity can be estimated using a number of approaches. The most general methods may utilize finite-difference schemes, where one samples pairs of points close by, and evaluates the difference. In neural networks, one may leverage (Weng et al., 2018b) frameworks such as PyTorch and TensorFlow to estimate generalized derivatives, usually Clarke subgradients. More recent methods specific to neural networks leverage the computation graph of the neural network and either integer programming (Tjeng et al., 2019; Anderson et al., 2020; Zhang et al., 2022; Schwan et al., 2023) with linear relaxations, or polynomial optimization (Chen et al., 2020; 2021) with SDP relaxations (Fazlyab et al., 2022;?; Wang et al., 2024; Yang et al., 2024; Pauli et al., 2024; Xu & Sivaranjani, 2024; Syed & Hu, 2025). (Xue et al., 2022; Pauli et al., 2024; Xu & Sivaranjani, 2024) focus on the decomposition of the large SDP variable utilizing some notion of sparsity. There are a variety of other approaches as well (Avant & Morgansen, 2024; Jordan & Dimakis, 2020; Bonaert et al., 2021; Jafarpour et al., 2022), including interval methods and branch and bound without convex relaxations (Bunel et al., 2018; Karg & Lucia, 2020; Wei & Kolter, 2022; Abad Rocamora et al., 2022), but their empirical performance is broadly speaking similar. Finally, Sridhar et al. (2022) develop the ideas of (Virmaux & Scaman, 2018; Goodfellow, 2018; Weng et al., 2018a) using ideas from extreme value theory.

Our work on the non-uniform sampling is also related (but not drawing on directly) to the Adaptive Sequential Elimination of (Aziz et al., 2018). Approaches such as (Jones et al., 1993; Gablonsky & Kelley, 2001) are also related, but not directly applicable, due to their assumptions of (global) Lipschitz properties.

More broadly, the estimation of robustness properties (Fazlyab et al., 2022) draws upon a long history of the study of neural networks in the control theory community (Psaltis et al., 1988; Parisini & Zoppoli, 1995, e.g.), and a long history of control-theoretic contributions to machine learning (Bunel et al., 2018, e.g.).

## 3 OUR APPROACH

We present a non-uniform stochastic sample approximation for estimating moduli of local continuity. First, we describe a straightforward stochastic approximation for the modulus of (local) Lipschitz continuity (Weng et al., 2018b) by

---

[1]The paper (Jordan & Dimakis, 2020) omits the assumption that $X$ is *convex*; although this assumption gets used in the proof of (Jordan & Dimakis, 2020, Theorem 1) when considering the function "$f(x + t(y - x))$" for arbitrary $x, y \in X$.

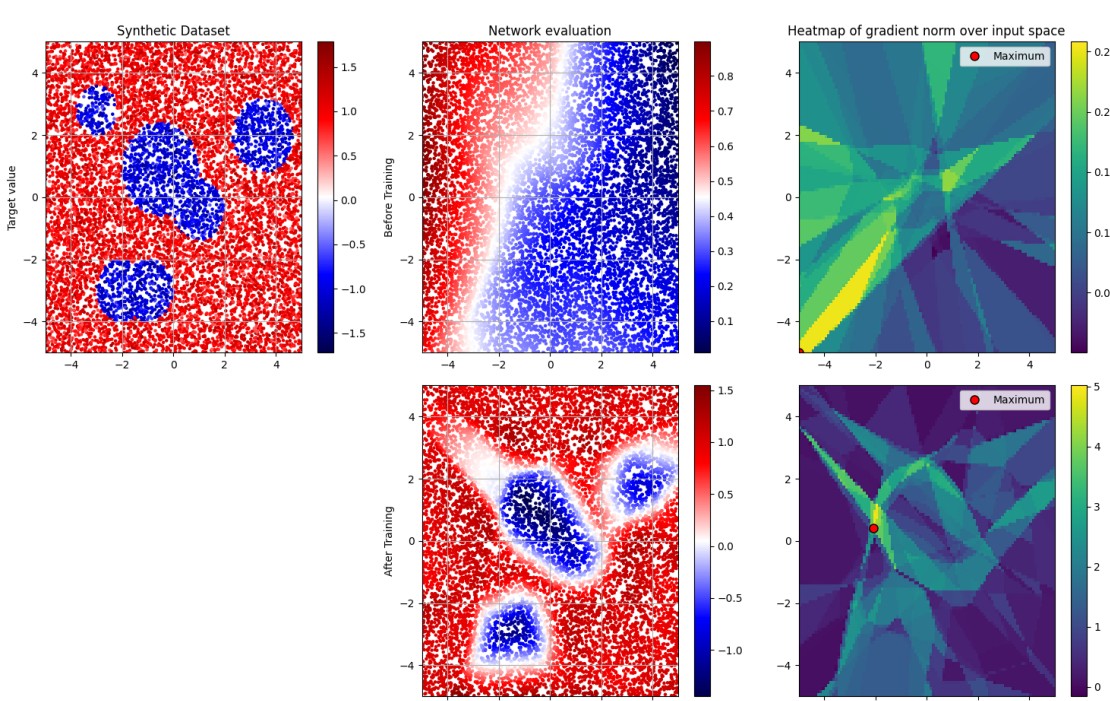

Figure 1: A running example. Left: training data. Middle: Neural network before (top) and after training (bottom). Right: Elements of Clarke subdifferential evaluated on the domain of the neural network prior to training (top) and after training (bottom). Maximum value sampled from the Clarke subdifferentials is marked with the red dot.

approximating the maximum norm of the (generalized) Jacobian matrix the neural network: $J_f(x)$. Subsequently, we extend this to non-uniform, adaptive sampling.

**Remark 4.** *Notice that many recent papers on the estimation of moduli of local continuity consider gradients and Jacobians, while they work with ReLU-based networks, which are non-smooth. In keeping with the literature, we will use $J_f(x)$ to denote the generalized Jacobian, computed using the Clarke subdifferential for modulus of local Lipschitz continuity, or some other generalized derivatives for other moduli of local continuity.*

### 3.1 SAMPLING CLARKE SUBDIFFERENTIAL

Neural networks are a special case of non-smooth, non-convex functions, known as functions definable in o-minimal structures (Ioffe, 2008). This class of functions comes with a chain rule for certain generalized derivatives, out of which the Clarke generalized derivative (Clarke, 1975) is the most popular, and stability of definability, which guarantees that a composition of definable functions remains definable. (See the supplementary material.) This enables commonly used frameworks such as PyTorch and TensorFlow to have a formula for all elementary activation functions and to apply automatic differentiation of the neural network by matrix multiplication. This way, one can evaluate elements of the Clarke subdifferential by a single call to the autograd library. By doing so for many inputs and looking at the maximum of the sampled values, one can estimate the modulus of local Lipschitz continuity. See Algorithm 1.

### 3.2 GENERALIZATIONS

One can generalize Algorithm 1 in two directions: first, to other moduli of local continuity, and second, to other forms of sampling. Algorithm 2 is parametrized by the modulus of local continuity and divides the $d$ dimensional input space into $k^d$ subregions and equally samples results from each one. This is illustrated in Figure 2. Notice that the change in

---
**Algorithm 1** Standard Stochastic Approximation, cf. (Weng et al., 2018b)
---
    **Input:** Neural network $f(x)$ and its convex open domain $D$, number of samples $N$.
    $r \leftarrow 0$
    **for** $1 \leq i \leq N$ **do**
        Select input point $x \in D$ uniformly at random from $D$
        Evaluate $f(x)$; compute $J_f(x)$ by backpropagation, cf. Remark 4
        Set $r \leftarrow \max\{r, \|J_f(x)\|_1\}$
    **end for**
    **return** $r$
---

sampling is not a very important one, yet: Experiments of Section 5 (esp. Figure 4) show that there is no particular correlation between the number of partitions and the precision of the approximation.

---
**Algorithm 2** Generalized Stochastic Approximation with Uniform Partitioning of the Domain
---
    **Input:** Norms $\|\cdot\|_\alpha$ and $\|\cdot\|_\beta$. Neural network $f(x)$ and its convex open domain $D$, number of samples $N$, number of divisions per dimensions $K$.
    $r \leftarrow 0$
    Partion domain $D$ into subregions $S \leftarrow \text{init\_subregions}(D, K)$
    **for** $s \in S$ **do**
        **for** $1 \leq i \leq N/|S|$ **do**
            Select input point $x \in s$ uniformly at random from subregion $s \subseteq D$.
            Evaluate $f(x)$; compute $\|J_f(x)\|_{\alpha,\beta}$ for a generalized Jacobian corresponding to the chosen notion of local continuity by backpropagation. Cf. Example 7.
            Set $r \leftarrow \max\{r, \|J_f(x)\|_{\alpha,\beta}\}$
        **end for**
    **end for**
    **return** $r$
---

### 3.3 NON-UNIFORM SAMPLING WITH UCB

Non-uniform sampling, including importance sampling (Kahn, 1950; Robert et al., 1999; Tokdar & Kass, 2010) and adaptive importance sampling (Bugallo et al., 2017), is a standard tool from statistics, which concentrates samples in regions where the estimate can be improved the most based on what has been sampled so far, without neglecting the other regions completely. In multi-armed bandit problems (Gittins et al., 2011; Lattimore & Szepesvári, 2020), the so-called upper-confidence bound policies (Lai, 1987; Auer et al., 2002) have a long history as means of non-uniform sampling that has been shown to be asymptotically optimal in a variety of setttings. The infinity-armed bandit problem (Berry et al., 1997; Wang et al., 2008) can be seen as an online sampling algorithm to estimate the maximum of a distribution. We suggest to see the estimation of a modulus of local continuity as an infinity-armed bandit problem (Berry et al., 1997; Wang et al., 2008), although without the commonly considered assumptions of Lipschitz continuity (Kleinberg et al., 2008; Bubeck et al., 2011), Bernoulli-distributed rewards (Berry et al., 1997; Gong & Sellke, 2023), and unique optimal arm (De Heide et al., 2021).

In particular, Algorithm 3 starts with an undivided domain $D$ of the neural network $f(x)$ as a single subregion. In each of $N$ iterations, one sample is randomly chosen from the subregion with the highest upper-confidence bound (also known as UCB score or an index), computed as:

$$\begin{cases} s_{max} + c \cdot \sqrt{\frac{\ln(t+1)}{s_n}} \cdot s_\sigma, & \text{if } s_n > 10 \\ \infty, & \text{if } s_n \leq 10 \end{cases}$$

where $s_{max}$ is the maximum encountered within the subregion so far, $s_n$ is the number of samples taken from the subregion so far, and $s_\sigma$ is the variance. After an exponentially increasing $(t_m, t_m^2, t_m^3, \ldots)$ number of iterations, a subregion that maximizes the UCB score equation 3.3 is subdivided. When the domain is polyhedral, the subregions are kept polyhedral, and subdivided by axis-aligned hyperplanes. In particular, when subdividing subregion $s$, its longest side is halved by a perpendicular, axis-aligned hyperplane. $s_n = 10$ threshold is set to get initial samples for each new subregion.

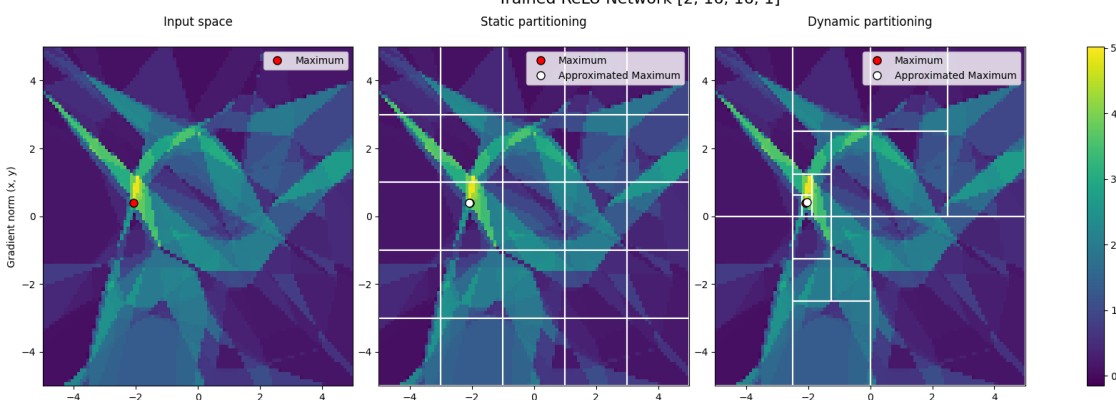

Figure 2: The running example continued. Left: Elements of Clarke subdifferential evaluated on the domain of the trained neural network. Middle: uniform partitioning of the domain. Right: non-uniform partitioning of the domain utilizing the upper-confidence bounds.

---

**Algorithm 3** Generalized Stochastic Approximation with Non-uniform Partitioning of the Domain

---

**Input:** Norms $\|\cdot\|_\alpha$ and $\|\cdot\|_\beta$. Neural network $f(x)$ and its convex open domain $D$, number of samples $N$, UCB exploration constant $c$, subdivision time multiplier $t_m$.

$r \leftarrow 0$

$t_{\text{subdivision}} \leftarrow t_m$

$S \leftarrow \{D\}$

**for** $1 \leq i \leq N$ **do**

    Select subregion $s \in S$ maximizing the UCB score equation 3.3 considering exploration constant $c$.

    **if** $i = t_{\text{subdivision}}$ **then**

        Replace $s \in S$ in $S$ with subdivide($s$)

        $t_{\text{subdivision}} \leftarrow t_{\text{subdivision}} \cdot t_m$

    **end if**

    Select input point $x \in s$ uniformly at random from subregion $s \subseteq D$.

    Evaluate $f(x)$; compute $\|J_f(x)\|_{\alpha,\beta}$ for a generalized Jacobian corresponding to the chosen notion of local continuity by backpropagation. Cf. Example 7.

    Set $r \leftarrow \max\{r, \|J_f(x)\|_{\alpha,\beta}\}$

**end for**

**return** $r$

---

## 4  AN ANALYSIS

From Theorem 3, it is clear by that Algorithms 1-3 will always return a lower bound for the Lipschitz constant, provided that "$J_f(x)$" is contained in the Clarke Jacobian $J_f^c(x)$. Our analysis focuses on the opportunities for replacing $J_f^c$ in Theorem 3 with some other set-valued generalized derivative

It has been observed many times (Davis et al., 2020; Bolte & Pauwels, 2021; Bareilles et al., 2025) that nearly all neural networks used in practice are definable in some o-minimal structure. Under this assumption, we describe in Theorem 6 below to what extent the role of the Clarke Jacobian in Theorem 3 can be replaced with a suitable alternative. See Appendix A for an elaboration of the current subsection, including a proof of Theorem 6.

We make the following assumptions:

- $\mathfrak{R} = (\mathbb{R}; <, +, \cdot, \dots)$ is an o-minimal expansion of the real field and "definable" means "definable in $\mathfrak{R}$ with parameters". The reader is free to focus on the special case where $\mathfrak{R} = (\mathbb{R}; <, +, \cdot)$ is the usual real field, in which case "definable" is a synonym for "semialgebraic"

- $\|\cdot\|_\alpha$ and $\|\cdot\|_\beta$ are norms on $\mathbb{R}^n$ and $\mathbb{R}^m$ respectively, not necessarily definable (in the sense of $\mathfrak{R}$)

- $X \subseteq \mathbb{R}^n$ is a definable open convex set
- $f : X \to \mathbb{R}^m$ is a definable $(\alpha, \beta)$-Lipschitz function

We introduce the following provisional terminology:

**Definition 5.** *We say a set-valued map $D : X \rightrightarrows \mathbb{R}^{m \times n}$ is **good** for $f$ if:*

- *$D(x) \subseteq J_f^c(x)$ for every $x \in X$, and*

- *there exists $X_0 \subseteq X$ such that:*
  - *$X \setminus X_0$ is a null set with respect to Lebesgue measure on $\mathbb{R}^n$, and*
  - *$D(x)$ is nonempty for every $x \in X_0$*

*[Note that we do not require $D$ or $X_0$ to be definable.]*

Under these assumptions, we have the following variation of Theorem 3:

**Theorem 6.** *If $D$ is good for $f$, then:*

$$L^{(\alpha,\beta)}(f, X) \;=\; \sup_{x \in X, G \in D(x)} \|G\|_{\alpha,\beta}$$

**Example 7.** *Apart from $D = J_f^c$, here are some examples of $D$ satisfying conditions of Theorem 6:*

- *$D$ can be the partially-defined single-valued map returning $J_f(x)$ at all points $x \in X$ for which $f$ is Fréchet-differentiable; more generally, given $r \geq 1$, $D$ can return $J_f(x)$ at all points $x \in X$ for which $f$ is $C^r$-smooth*

- *$D$ can be an arbitrary selection of $J_f^c$ defined a.e. in $X$*

- *when $m = 1$, $D$ can be the Fréchet, limiting, or Clarke subdifferential; e.g., (Bolte et al., 2007).*

Note that Theorem 6 can fail if we do not assume that $f$ is definable:

**Example 8.** *Let $f : \mathbb{R} \to \mathbb{R}$ be a Lipschitz function with Lipschitz constant 1 such that $J_f^c(x) = [-1, 1]$ for every $x \in \mathbb{R}$; such functions occur generically according to (Ioffe, 2008; Borwein & Wang, 1997) although they cannot be definable (as this would violate the Smooth Monotonicity Theorem (Van den Dries, 1998, 7.2.5)). Set $D(x) := \{0\}$ for every $x \in \mathbb{R}$; then $D$ is good for $f$, however $\sup_{x \in \mathbb{R}, G \in D(x)} \|G\|_{\alpha,\beta} = 0 \neq 1$.*

As our idealized model for "$J_f(x)$" gets closer to how automatic differentiation (AD) is actually implemented by PyTorch or TensorFlow, then Theorem 6 may no longer be true due to sporadic behavior on null sets:

**Remark 9.** *Suppose $D$ is either (1) the output of AD as modeled by selection Jacobians (Bolte & Pauwels, 2020), or (2) a conservative mapping for $f$ (Bolte & Pauwels, 2021). Then $D = \{J_f(x)\}$ a.e. on $X$ and so $L^{(\alpha,\beta)}(f, X) \leq \sup_{x \in X, G \in D(x)} \|G\|_{\alpha,\beta}$. However, equality can fail in this case as (Bolte & Pauwels, 2020; 2021) provide easy examples where $f \equiv 0$, although $D(x)$ can contain nonzero vectors.*

The consistency of the estimate obtained by Algorithm 3 is clear from the fact that we are sampling from any point in the domain $D$ with a positive probability and the strong law of large numbers (Robert et al., 1999, cf. Theorem 4.7.3). The estimate is not unbiased, but only asymptotically unbiased (Robert et al., 1999). The asymptotic optimality of Algorithm 3 within a certain class of sampling schemes is less trivial, but follows from well-known analyses (Munos, 2011) of UCB policies for the infinity-armed bandit problems.

## 5 EXPERIMENTAL RESULTS

As a proof of concept, we have implemented our approach in PyTorch using their automatic differentiation engine (torch.autograd). In our experiments, we have used ReLU-based neural nets (composed of torch.nn.Linear and torch.nn.ReLU layers). To describe their dimensions, we use the notation such as of $[2, w_1, w_2, 1]$, where there are two

scalars on the inputs, a linear layer with $w_1$ output features, ReLU activation, a linear layer with $w_1$ input features and $w_2$ output features, ReLU activation, and a linear layer with $w_2$ input features and 1 output feature. For the purposes of visualization, we often focus on the two-dimensional input. We generate synthetic datasets to match, one of which is displayed on the left in Figure 1, with points randomly generated hyperspheres with points generated randomly with mean $-1.0$, while the remainder of the hypercube is filled with values generated at random with mean 1.0.

In Figures 1–3, we focus on the running-example neural network [2, 16, 16, 1] presented in Figure 1. In Figure 4 (on the right), we summarize the experiments where all the generated datasets had 3 hyperspheres of random radius $0.1 \times r_{domain} \leq r_{sphere} \leq 0.4 \times r_{domain}$ randomly centered on the input space. For each number of hidden layers in the right plot in Figure 4, 30 test runs were evaluated, for each test run a new neural net was trained on a random data set with 800 points. Training was performed with torch.nn.MSELoss() as loss function, torch.optim.Adam() as optimizer with learning rate $5 \times 10^{-4}$ for 500 epochs. All 3 algorithms were limited to 60000 evaluations, For Algorithm 2, 2 divisions per dimension were set, resulting in $2^7$ subregions; for Algorithm 3 the exploration parameter $c$ (3.3) was set to 10 and the subdivision time multiplier was set to 2.

In Figure 4 (on the left), we ran Algorithm 2 with different partition parameter on the one trained neural net of depth 4 and width 8, for each parameter value 100 runs were evaluated. The 2D heat maps of the gradients that are present in the figures were generated by evaluating the $400^2$ grid. The red dot in the plots presented as Maximum is within the $0.001\%$ error of the actual value computed by LipMIP. In Figure 5, we evaluate using untrained neural nets. Stochastic Approximation and LipSDP were run without parallelization, and 5 cores were allocated for Gurobi engine in LipMIP.

In Table 1, we evaluate using neural nets trained on binary MNIST (classification between 1 and 7), following the benchmarking procedure suggested by (Tjeng et al., 2019). Column "setup" presents the dimensions of the net, and acceptable error of Algorithms 1 and 3 (when applicable). For each net, we ran 5 evaluations on trained net and best result in terms of relative error for each method was chosen. For the last two experiments, run time of LipMIP (Tjeng et al., 2019) was limited. Considering that LipSDP produces an upper bound, the lowest (best) result is presented. For Algorithms 1 and 3, the highest result is presented. Rows 2 and 4 present experiments, where Algorithms 1 and 3 could compare the outcomes against the LipMIP result. (This is of interest, because one may often have many similar nets and may wish to estimate of their moduli of continuity one by one, using as few iterations as possible.) Because LipMIP computes the modulus of $\langle c, f \rangle$, modified versions of Algorithm 3 were used, where the modulus of local continuity was approximated from $\langle c, f \rangle$, rather than from the whole output of $f$. Rows 6 and 7 present experiments, where the scalability of Algorithm 3 strictly exceeds the scalability of LipMIP. Rows 6 and 7 present experiments, where the scalability of Algorithm 3 strictly exceeds the scalability of LipMIP. By "?" in the "Relative error" column, we indicate that we do not have the exact modulus of local continuity. The comparison against LipSDP shows that our estimates improve upon LipSDP and its variants, whose upper bound is very loose (at least 356% higher in row 7), without requiring a longer run-time.

## 6    CONCLUSIONS AND LIMITATIONS

We have elaborated upon the connections between "generalized derivatives" and estimating Lipschitz constants in the setting of *tame optimization* (Ioffe, 2008; Bareilles et al., 2025). We have also presented non-uniform sampling within the context of estimating the modulus of continuity using, and showed that it improves the scalability of the stochastic sample approximations therein, improving also over some well-established methods not based on stochastic sample approximations (LipMIP, LipSDP). The advantage of non-uniform sampling (Algorithm 3) is clear especially when there are both areas of vanishing (sub)gradients (Hochreiter, 1998) and areas with high variance of the subgradients in the domain. This is common (Hanin, 2018) such as in the case of deeper and wider nets.

A key limitation of our method, as well as any other method for the verification of properties of neural networks known presently, is scalability. While our algorithms can be applied to a neural network in any dimension, and our results (e.g., Table 1 and Figure 5) suggest some improvement in run-time for instances where all methods are applicable, this is far from the scale of foundational models utilized in many applications. One may wish to bound the rates of convergence of the UCB-based policies and to improve the per-iteration run-time. In order to improve the per-iteration run-time (the number of sample paths per second), one may wish to implement the approach in JAX or Enzyme (Moses & Churavy, 2020).

The performance of the Algorithm 3 is also affected by the choice of the UCB exploration parameter $c$; see Appendix B for a discussion. Related methods have extensive applications, e.g., in the safety filters for AI systems, and in the design of controllers for such closed loops with guarantees of ergodic behaviour.

Table 1: Comparison of methods on Binary MNIST following the benchmarking procedure suggested in the LipMIP paper (Tjeng et al., 2019).

| Setup | Method | Value | Time (s) | Relative Error (%) |
|---|---|---|---|---|
| [784, 8, 2] | LipMIP | 77.08519 | 1.523 | 0 |
| | LipSDP | 100.76831 | 7.610 | +30.7 |
| | Algorithm 1 (5000) | 77.07964 | 0.312 | -0.007 |
| | Algorithm 3 (5000) | 77.07964 | 1.330 | -0.007 |
| [784, 8, 2] Approximation tolerance 1% | LipMIP | 77.08519 | 1.523 | 0 |
| | LipSDP | 100.76831 | 7.610 | +30.7 |
| | Algorithm 1 (5000) | 77.07964 | 0.0132 | -0.007 |
| | Algorithm 3 (5000) | 77.07964 | 0.0004 | -0.007 |
| [784, 8, 8, 2] | LipMIP | 79.24930 | 6.214 | 0 |
| | LipSDP | 104.73195 | 8.987 | +32.155 |
| | Algorithm 1 (10000) | 79.01881 | 0.776 | -0.291 |
| | Algorithm 3 (10000) | 79.01880 | 2.696 | -0.291 |
| [784, 8, 8, 2] Approximation tolerance 1% | LipMIP | 79.24930 | 6.214 | 0 |
| | LipSDP | 104.73195 | 8.987 | +32.155 |
| | Algorithm 1 (10000) | 79.01881 | 0.051 | -0.291 |
| | Algorithm 3 (10000) | 79.01880 | 0.408 | -0.291 |
| [784, 20, 20, 2] | LipMIP | 349.67096 | 6.048 | 0 |
| | LipSDP | 479.54083 | 12.48653 | +37.140 |
| | Algorithm 1 (150000) | 277.22370 | 11.931 | -20.719 |
| | Algorithm 3 (150000) | 290.12729 | 44.975 | -17.028 |
| [784, 8, 8, 8, 2] | LipMIP (timeout) | 114.48170 | 120.0 | ? |
| | LipSDP | 125.98295 | 8.987 | ? |
| | Algorithm 1 (100000) | 89.87323 | 9.255 | ? |
| | Algorithm 3 (100000) | 89.87323 | 20.522 | ? |
| [784, 8, 8, 8, 8, 8, 2] | LipMIP (timeout) | 519.65584 | 300.0 | ? |
| | LipSDP | 429.07478 | 11.472 | ? |
| | Algorithm 1 (100000) | 94.14903 | 12.417 | ? |
| | Algorithm 3 (100000) | 94.14903 | 24.549 | ? |

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

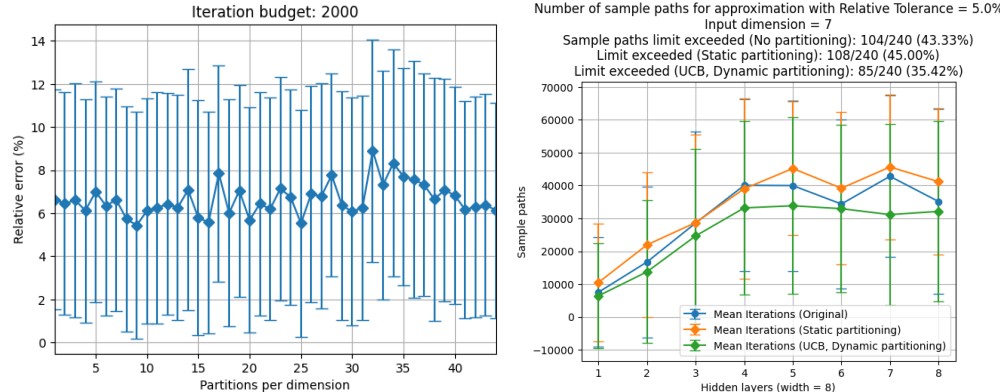

Figure 4: Illustrative results. Left: relative error of Algorithm 2 is robust with respect to the number of subregions. Right: performance of Algorithms 1 and 2 is very similar, when considering one and the same subdiffrential. Algorithm 3 strictly improves upon both Algorithms 1 and 2.

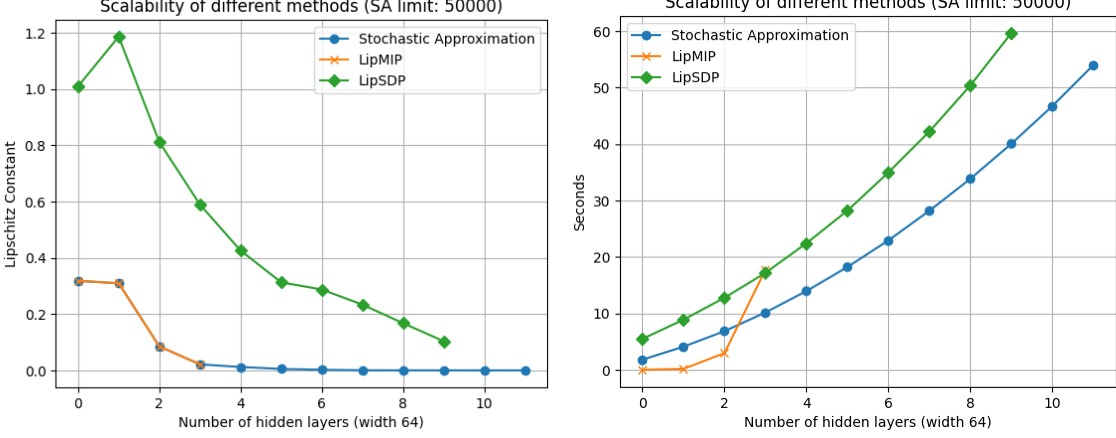

Figure 5: Illustrative comparison against LipMIP and LipSDP. Left: with the number of hidden layers (each of width 64), the modulus of local Lipschitz continuity decreases sharply. LipSDP overestimates the modulus by a substantial margin, while stochastic approximation with 50000 samples tracks the true values computed using LipMIP. Right: within a given run-time limit of 60 seconds, Algorithm 3 with 50000 samples scales to depth-11 networks (704 neurons), where LipSDP utilizing Mosek scales to depth 9 (576 neurons) and LipMIP utilizing Gurobi scales to depth 3 (192 neurons).

# A  TECHNICAL APPENDICES AND SUPPLEMENTARY MATERIAL

In this appendix we give a proof of Theorem 6 (Theorem 41 below) which is an o-minimal variant of (Jordan & Dimakis, 2020, Theorem 1) (Theorem 3 above). Theorem 6 follows from the following nontrivial but well-known facts about functions $f : \mathbb{R}^n \to \mathbb{R}^m$:

1. if $f$ is definable in an o-minimal expansion of the real field (in short: $f$ *is definable*), then $f$ is $C^1$ almost everywhere (cf. Lemma 24)

2. (Rademacher's Theorem) if $f$ is locally Lipschitz, then $f$ is Fréchet-dfferentiable almost everywhere (cf. Lemma 29)

3. (Warga's Theorem) for locally Lipschitz $f$, the Clarke Jacobian $J_f^c$ of $f$ is "blind" to sets of measure zero (cf. Lemma 31)

Taking these facts for granted, here we otherwise give a self-contained account in order to see where the assumption of definability is relevant and where it is not. In particular, via a Path Lemma 32 we are able to avoid the Lebesgue integral in favor of the Riemann integral in the proof of (Jordan & Dimakis, 2020, Theorem 1) (Corollary 34 below) in the case that $f$ is definable.

We set the following assumptions/conventions throughout the appendix (more assumptions will be added along the way):

- $\mathfrak{R} = (\mathbb{R}; <, +, \cdot, \ldots)$ is an o-minimal expansion of the real field and "definable" means "definable in $\mathfrak{R}$ with parameters"

- we assume the reader is familiar with the concepts of *o-minimality* and *definability*; our main references for these concepts include (Van den Dries, 1998) and (Aschenbrenner et al., 2017, Appendix B). The reader is free to consider the special case where $\mathfrak{R} = (\mathbb{R}; <, +, \cdot)$ is the real field, and thus "definable" means "semialgebraic"

- $f : \mathbb{R}^n \rightharpoonup \mathbb{R}^m$ is a partial function

- $X \subseteq \mathrm{dom}(f) \subseteq \mathbb{R}^n$

- for points $x, y \in \mathbb{R}^n$, we denote the line segment from $x$ to $y$ by:

$$[x, y] := \{(1 - t)x + ty : t \in [0, 1]\}$$

- recall (Rockafellar & Wets, 2009, Chapter 5) that a *set-valued map* $D : A \rightrightarrows B$ is by definition a function $D : A \to \mathcal{P}(B)$ (where $\mathcal{P}(B)$ denotes the powerset of $B$). When convenient, we identify $D$ with its *graph* $\mathrm{gph}\, D := \{(a, b) : b \in D(a)\} \subseteq A \times B$. If $A \subseteq \mathbb{R}^m$ and $B \subseteq \mathbb{R}^n$, then we say $D$ is *definable* if $\mathrm{gph}\, D \subseteq \mathbb{R}^{m+n}$ is definable.

- we roughly follow the notation and definitions as in (Jordan & Dimakis, 2020)

NORMS ON VECTOR SPACES OVER $\mathbb{R}$

The concept of *Lipschitz constant* is relative to a choice of norms on the spaces $\mathbb{R}^m, \mathbb{R}^n$. Here we recall some basic definitions and properties concerning norms.

**Definition 10.** *Suppose $V$ is a vector space over $\mathbb{R}$. A **norm** $\| \cdot \|$ on $V$ is a function:*

$$\| \cdot \| : V \to [0, +\infty)$$

*which satisfies for all $x, y \in V$ and $\lambda \in \mathbb{R}$:*

1. *(Positive definiteness) $\|x\| = 0$ iff $x = 0$*

2. *(Absolute homogeneity) $\|\lambda x\| = |\lambda| \|x\|$*

3. *(Triangle inequality) $\|x + y\| \leq \|x\| + \|y\|$*

**Lemma 11.** *Suppose $\| \cdot \|$ is a norm on $\mathbb{R}^n$.*

*1. If $x_i$ is a sequence in $\mathbb{R}^n$ such that $x_i \to x$, then:*

$$\|x\| = \lim_{i\to\infty} \|x_i\| \le \sup\{\|x_i\| : i \ge 0\}$$

*2. If $X \subseteq \mathbb{R}^n$, then:*

$$\sup_{x\in X} \|x\| = \sup_{x\in\mathrm{conv}(X)} \|x\|$$

*Proof.* (1) Follows from the fact that $\|\cdot\| : \mathbb{R}^n \to \mathbb{R}$ is a continuous function.

(2) The direction "$\le$" is clear. For the "$\ge$" direction, suppose $x \in \mathrm{conv}(X)$, then there is $k \ge 1$ and $x_1, \ldots, x_k \in X$ and $\lambda_1, \ldots, \lambda_k \in [0,1]$ such that $x = \sum_{1\le i\le k} \lambda_i x_i$ and $\sum_{1\le i\le k} \lambda_i = 1$. Suppose $i_0 \in \{1, \ldots, k\}$ satisfies $\|x_{i_0}\| = \max\{\|x_i\| : i = 1, \ldots, k\}$. Then:

$$\|\textstyle\sum_{1\le i\le k} \lambda_i x_i\| \le \sum_{1\le i\le k} \lambda_i \|x_i\| \le \sum_{1\le i\le k} \lambda_i \|x_{i_0}\| = \|x_{i_0}\| \qquad \square$$

The following lemma includes a template for the type of construction which occurs in the definition of *Clarke Jacobian*:

**Lemma 12.** *Given $S \subseteq X$ and $J : S \to \mathbb{R}^m$, define:*

$$D_J : X \rightrightarrows \mathbb{R}^m, \quad x \mapsto D_J(x) := \{\lim_{i\to\infty} J(x_i) : x_i \to x, x_i \in S\}$$

*Then:*

$$\sup_{x\in S} \|J(x)\|_\beta = \sup_{x\in X, G\in\mathrm{conv}(D_J(x))} \|G\|_\beta$$

*Proof.* It is clear that the direction "$\le$" holds. Conversely, first note that by Lemma 11(2) we have:

$$\sup_{x\in X, G\in\mathrm{conv}(D_J(x))} \|G\|_\beta = \sup_{x\in X} \sup_{G\in\mathrm{conv}(D_J(x))} \|G\|_\beta = \sup_{x\in X} \sup_{G\in D_J(x)} \|G\|_\beta$$

Next, let $x \in X$ and $G \in D_J(x)$ be arbitrary, and take $x_i \in S$ such that $x_i \to x$ and $J(x_i) \to G$. Then by Lemma 11(1) we have

$$\|G\|_\beta \le \sup\{\|J(x_i)\|_\beta : i \ge 0\} \le \sup_{x\in S} \|J(x)\|_\beta$$

Since $x, G$ were arbitrary, this yields the "$\ge$" direction. $\qquad\square$

As is common, we shall identify $\mathbb{R}^n$ with its dual space. A norm $\|\cdot\|$ on $\mathbb{R}^n$ induces a **dual norm** $\|\cdot\|_*$ on the dual space $\mathbb{R}^n$:

$$\|\cdot\|_* : \mathbb{R}^n \to [0, +\infty), \quad y \mapsto \|y\|_* := \sup_{\|x\|\le 1} \langle x, y\rangle$$

In our finite-dimensional setting we have $\|\cdot\|_{**} = \|\cdot\|$. We will also make use of Hölder's inequality for dual norms:

**Lemma 13.** *(Jordan & Dimakis, 2020, Proposition 1) Suppose $\|\cdot\|$ is a norm on $\mathbb{R}^n$. Then for every $x, y \in \mathbb{R}^n$ we have:*

$$\langle x, y\rangle \le \|x\| \cdot \|y\|_*$$

Using the double dual norm and Hölder's inequality, we get:

**Lemma 14** (Continuous triangle inequality). *Suppose $g : [0,1] \to \mathbb{R}^m$ is Riemann integrable and $\|\cdot\|$ is a norm on $\mathbb{R}^m$. Then $\|g\| : [0,1] \to \mathbb{R}$ is Riemann integrable and:*

$$\|\textstyle\int_0^1 g(t)dt\| \le \int_0^1 \|g(t)\|dt$$

*Proof.* The first statement is clear. Using the double dual norm and linearity, it suffices to show:

$$\|\textstyle\int_0^1 g(t)dt\| = \sup_{\|y\|_*\le 1}\langle y, \textstyle\int_0^1 g(t)dt\rangle = \sup_{\|y\|_*\le 1}\textstyle\int_0^1 \langle y, g(t)\rangle dt \le \int_0^1 \|g(t)\|dt$$

Let $y$ be arbitrary such that $\|y\|_* \le 1$, then it suffices to show:

$$\textstyle\int_0^1 \langle y, g(t)\rangle dt \le \int_0^1 \|g(t)\|dt$$

For this, it suffices to show for every $t \in [0, 1]$:

$$\langle y, g(t) \rangle \ \leq \ \|g(t)\|$$

However this follows from Hölder's inequality 13 for dual norms:

$$\langle y, g(t) \rangle \ \leq \ \|y\|_* \|g(t)\| \ \leq \ \|g(t)\|$$

since $\|y\|_* \leq 1$. $\square$

For the rest of this appendix, we further assume:

- $\| \cdot \|_\alpha$ and $\| \cdot \|_\beta$ are norms on $\mathbb{R}^n$ and $\mathbb{R}^m$ respectively (we don't require these norms to be definable, e.g., the graph of the function $\| \cdot \|_\alpha : \mathbb{R}^n \to \mathbb{R}$, as a subset of $\mathbb{R}^{n+1}$, is not assumed to be definable in $\mathfrak{R}$, likewise for $\| \cdot \|_\beta$)

We may also define the **matrix norm** $\| \cdot \|_{\alpha,\beta}$ on $\mathbb{R}^{m \times n}$ induced by $\| \cdot \|_\alpha$ and $\| \cdot \|_\beta$:

$$\| \cdot \|_{\alpha,\beta} \ : \ \mathbb{R}^{m \times n} \to [0, +\infty), \quad A \ \mapsto \ \sup_{\|x\|_\alpha \leq 1} \|Ax\|_\beta$$

There is also an analogous Hölder-type inequality for matrix norms:

**Lemma 15.** *(Jordan & Dimakis, 2020, Proposition A.2) For every $A \in \mathbb{R}^{m \times n}$ and $x \in \mathbb{R}^n$ we have:*

$$\|Ax\|_\beta \ \leq \ \|A\|_{\alpha,\beta} \|x\|_\alpha$$

THE LIPSCHITZ PROPERTY

**Definition 16.** *(Jordan & Dimakis, 2020, Definition 1) The **local** $(\alpha, \beta)$-**Lipschitz constant** of $f$ over $X$ is the (possibly infinite) quantity:*

$$L^{(\alpha,\beta)}(f, X) \ := \ \sup_{x,y \in X} \frac{\|f(x) - f(y)\|_\beta}{\|x - y\|_\alpha} \quad (x \neq y)$$

*Moreover, if $L^{(\alpha,\beta)}(f, X)$ is finite, we say that $f$ is $(\alpha, \beta)$-**Lipschitz** over $X$. We say that $f$ is **locally** $(\alpha, \beta)$-**Lipschitz** over $X$ if for every $x \in X$, there exists a neighbourhood $U$ of $x$ such that $f$ is $(\alpha, \beta)$-Lipschitz over $X \cap U$.*

The following is well-known:

**Lemma 17.** *If $X$ is compact and $f$ is locally $(\alpha, \beta)$-Lipschitz over $X$, then $f$ is $(\alpha, \beta)$-Lipschitz over $X$.*

DIRECTIONALLY DIFFERENTIABLE, FRÉCHET DIFFERENTIABLE, AND $C^1$

For the rest of this appendix, we further assume:

- $X$ is open

**Definition 18.** *Given $x \in X$ and $v \in \mathbb{R}^n$, we say that $f$ is **directionally differentiable** at $x$ in the direction $v$ if there exists $\ell \in \mathbb{R}^m$ such that:*

$$\lim_{t \downarrow 0} (f(x + tv) - f(x))/t \ = \ \ell$$

*in which case we define the **directional derivative** $f'(x; v)$ of $f$ at $x$ in the direction $v$ to be $f'(x; v) := \ell$. We say that $f$ is **directionally differentiable** at $x$ if $f$ is directionally differentiable at $x$ in the direction $v$ for every $v \in \mathbb{R}^n$.*

Here is the basic relationship between a directional derivative and the Lipschitz constant:

**Lemma 19.** *If $f$ is directionally differentiable at $x$ in the direction $v$, then:*

$$\|f'(x; v)\|_\beta \ \leq \ L^{(\alpha,\beta)}(f, X) \|v\|_\alpha$$

*Proof.* We may suppose $v \neq 0$. For every $\varepsilon > 0$ we can find $t > 0$ sufficiently small such that $\|f'(x; v)\|_\beta$ is within $\varepsilon$ from the quantity:

$$\frac{\|f(x + tv) - f(x)\|_\beta}{t} = \|v\|_\alpha \frac{\|f(x + tv) - f(x)\|_\beta}{\|(x + tv) - x\|_\alpha}$$

Since the quantity on the righthand side contributes to the definition of $L^{(\alpha,\beta)}(f, X)\|v\|_\alpha$ as a certain supremum, the inequality follows. $\qquad\square$

**Definition 20.** *We say that $f$ is **Fréchet-differentiable** at $x$ if there exists a linear map $L : \mathbb{R}^n \to \mathbb{R}^m$ such that $f(x) + L(y - x)$ is a first-order approximation of $f$ at $x$, i.e., we have:*

$$\lim_{y \to x} \frac{\|f(y) - f(x) - L(y - x)\|_\beta}{\|y - x\|_\alpha} = 0$$

*in which case we define the **Jacobian** $J_f(x)$ of $f$ at $x$ to be the (necessarily unique) linear map $J_f(x) = L$; we shall identify the Jacobian $J_f(x)$ with a matrix in $\mathbb{R}^{m \times n}$ which represents $L$ with respect to the standard basis.*

**Lemma 21.** *If $f$ is Fréchet-differentiable at $x$, then $f$ is directionally differentiable at $x$ and for every $v \in \mathbb{R}^n$, we have:*

$$J_f(x)v = f'(x; v)$$

Here is a relationship between the Jacobian at a point and the Lipschitz constant:

**Lemma 22.** *If $f$ is Fréchet-differentiable at $z \in X$, then:*

$$\|J_f(z)\|_{\alpha,\beta} \leq L^{(\alpha,\beta)}(f, X)$$

*Proof.* By Lemmas 21 and 19 we have:

$$\|J_f(z)\|_{\alpha,\beta} = \sup_{\|v\|_\alpha \leq 1} \|J_f(z)v\|_\beta = \sup_{\|v\|_\alpha \leq 1} \|f'(z; v)\|_\beta \leq \sup_{\|v\|_\alpha \leq 1} L^{(\alpha,\beta)}(f, X)\|v\|_\alpha = L^{(\alpha,\beta)}(f, X) \qquad \square$$

**Definition 23.** *Given $x \in X$, we say that $f$ is $C^1$ at $x$ if there exists an open neighbourhood $U$ of $x$ in $X$ such that $f$ is Fréchet-differentiable at each $y \in U$ and moreover, the function $y \mapsto J_f(y) : U \to \mathbb{R}^{m \times n}$ is continuous.*

We let $\mathrm{Diff}(f) \subseteq X$ denote the set of points $x \in X$ such that $f$ is Fréchet-differentiable at $x$ and we let $\mathrm{Diff}^1(f)$ denote the set of points $x \in X$ such that $f$ is $C^1$ at $x$. Clearly $\mathrm{Diff}^1(f) \subseteq \mathrm{Diff}(f)$.

Here is the main fact we will use about definable functions:

**Lemma 24.** *If $f : X \to \mathbb{R}^m$ is definable, then:*

 1. $\mathrm{Diff}(f)$ *and* $\mathrm{Diff}^1(f)$ *are definable*

 2. $\dim(X \setminus \mathrm{Diff}^1(f)) < n$, *and thus* $X \setminus \mathrm{Diff}^1(f)$ *is nowhere dense and has Lebesgue measure zero*

*Remark about proof.* (1) is an easy exercise in definability. (2) is nontrivial and follows from *Smooth Cell Decomposition* (Van den Dries, 1998, 7.3.2). Here the dimension $\dim$ is taken in the sense of definable sets in an o-minimal structure (Van den Dries, 1998, 4.1) which agrees with and generalizes the usual notion of *dimension* for $C^1$ manifolds, at least in the case that the manifold is presented as an embedded submanifold of $\mathbb{R}^n$ and the underlying set of the manifold is a definable subset of $\mathbb{R}^n$. The claim about Lebesgue measure follows from the fact that for any connected embedded $C^1$ submanifold $M \subseteq \mathbb{R}^n$ (definable or not), if $\dim M < n$, then $M$ has Lebesgue measure zero. $\qquad\square$

A SUPREMUM OF JACOBIANS BOUND A DIFFERENCE QUOTIENT

The lemma here is routine although we show how to use the Riemann integral instead of the Lebesgue integral when we are in the definable setting. Here is the setup:

 • Fix distinct points $x, y \in \mathbb{R}^n$ such that $[x, y] \subseteq X$
 • let $L := [x, y] \setminus \{x, y\}$ be the "open" line segment from $x$ to $y$

- $f : X \to \mathbb{R}^m$ is $(\alpha, \beta)$-Lipschitz

- $f$ is either $C^1$ on $L$, or definable and Fréchet-differentiable on $L$

**Lemma 25.** *In the above setup we have:*

$$\|f(y) - f(x)\|_\beta \ \leq \ \sup_{z \in L} \|J_f(z)\|_{\alpha,\beta} \cdot \|y - x\|_\alpha$$

*Proof.* First define the function:

$$h \ : \ [0,1] \to \mathbb{R}^m, \quad t \ \mapsto \ f((1-t)x + ty)$$

We know that:

- $h$ is continuous

- $h$ is differentiable on $(0,1)$ with derivative $h'(t) = J_f((1-t)x + ty)(y - x)$, computed via the chain-rule for Fréchet-differentiable functions

- in particular, by Lemma 22, $h'$ is bounded because $f$ is Lipschitz over the compact set $[x, y]$; c.f. Lemma 17

- on $(0,1)$, $h'$ is continuous at all but finitely many points: by assumption either $f$ is $C^1$, or if $f$ is definable then this follows by the *Monotonicity Theorem* (Van den Dries, 1998, 3.1.2) (in fact, $h'$ is actually continuous on $(0,1)$ by (Fischer, 2005, 5.7))

Next, define the function:

$$g \ : \ [0,1] \to \mathbb{R}^m, \quad t \ \mapsto \ g(t) \ := \ \begin{cases} J_f((1-t)x + ty)(y - x) & \text{if } t \in (0,1) \\ 0 & \text{if } t = 0, 1 \end{cases}$$

Then $g$ is bounded and continuous at all but finitely many points, hence $g$ is Riemann integrable. Moreover, $h'(t) = g(t)$ on $(0,1)$ and so by the *Fundamental Theorem of Calculus* (Ross, 2013, 34.1) we have:

$$f(y) - f(x) \ = \ h(1) - h(0) \ = \ \int_0^1 g(t)dt$$

Next, note that we have the following bound on $g(t)$, for $t \in (0,1)$, by Lemma 15:

$$\|g(t)\|_\beta \ = \ \|J_f((1-t)x + ty)(y - x)\|_\beta \ \leq \ \|J_f((1-t)x + ty)\|_{\alpha,\beta}\|y - x\|_\alpha \ \leq \ \sup_{z \in L} \|J_f(z)\|_{\alpha,\beta} \cdot \|y - x\|_\alpha$$

Moreover, the overall bound holds for all $t \in [0,1]$. The main inequality now proceeds as follows:

$$\begin{aligned}
\|f(y) - f(x)\|_\beta \ &= \ \|\int_0^1 g(t)dt\|_\beta \\
&\leq \ \int_0^1 \|g(t)\|_\beta dt \quad \text{by Lemma 14} \\
&\leq \ \int_0^1 \sup_{z \in L} \|J_f(z)\|_{\alpha,\beta} \cdot \|y - x\|_\alpha dt \\
&= \ \sup_{z \in L} \|J_f(z)\|_{\alpha,\beta} \cdot \|y - x\|_\alpha \qquad \square
\end{aligned}$$

The following is now immediate from Lemmas 22 and 25:

**Corollary 26** (Lemma 2)**.** *If $f : \mathbb{R}^n \to \mathbb{R}^m$ is $C^1$ and $(\alpha, \beta)$-Lipschitz over an open convex set $X$, then:*

$$L^{(\alpha,\beta)}(f, X) \ = \ \sup_{x \in X} \|J_f(x)\|_{\alpha,\beta}$$

In the case that $f$ is definable, then we may weaken the $C^1$ assumption in Corollary 26 to Fréchet-differentiable and the same argument works; although in this case, this statement will be superseded by Lemma 33 below.

THE CLARKE JACOBIAN

In this subsection we assume:

- $f : X \to \mathbb{R}^m$ is locally $(\alpha, \beta)$-Lipschitz

**Definition 27.** *(Clarke, 1990, §2.6) Define the **Clarke Jacobian** $J_f^c$ of $f$ over $X$ to be the set-valued map:*

$$J_f^c : X \rightrightarrows \mathbb{R}^{m \times n}, \quad z \mapsto J_f^c(z) := \operatorname{conv}\{\lim_{j \to \infty} J_f(z_j) : z_j \to z, z_j \in \operatorname{Diff}(f)\}$$

The following is immediate from the definition:

**Lemma 28.** *If $x \in \operatorname{Diff}(f)$, then $\{J_f(x)\} \subseteq J_f^c(x)$; moreover, if $f$ is $C^1$ at $x$ then equality holds.*

Note that if $f$ is definable, then Lemma 24 guarantees that $\dim(X \setminus \operatorname{Diff}(X)) < n$ and thus $X \setminus \operatorname{Diff}(f)$ has Lebesgue measure zero. The general case is handled by:

**Lemma 29** (Rademacher). *(Rademacher, 1919) The set $X \setminus \operatorname{Diff}(f) \subseteq \mathbb{R}^n$ has Lebesgue measure zero.*

**Lemma 30.** *For every $x \in X$, $J_f^c(x)$ is nonempty, compact, and convex.*

*Proof.* Convexity is clear and compactness follows from the fact that lipschitz functions have bounded Jacobian (Lemma 22). Nonemptiness follows from Rademacher's Theorem. □

It is also convenient to consider the following "variant" of $J_f^c$; suppose $N \subseteq X$ has Lebesgue measure zero and define:

$$J_f^{c,N} : X \rightrightarrows \mathbb{R}^{m \times n}, \quad z \mapsto J_f^{c,N}(z) := \operatorname{conv}\{\lim_{j \to \infty} J_f^c(z_j) : z_j \to z, z_j \in \operatorname{Diff}(f) \setminus N\}$$

**Lemma 31** (Warga). *(Warga, 1981, Theorem 4) For any set $N \subseteq X$ of Lebesgue measure zero, we have $J_f^{c,N} = J_f^c$.*

**Exercise.** *Assuming $f$ and $N$ are definable, give a proof of Lemma 31 which does not use measure theory (use the fact that $\dim N < n$).*

THE PATH LEMMA

Here is the setup:

- $X \subseteq \mathbb{R}^n$ is a definable open convex set

**Lemma 32** (Path Lemma). *Suppose $x, y \in X$ and $B \subseteq \mathbb{R}^n$ is a definable set such that $\dim B < n$. For every $\varepsilon > 0$, there exists $k \geq 0$ and points $x_0 = x, x_1, \ldots, x_k, x_{k+1} = y$ in $X$ such that:*

- *$\sum_{0 \leq i \leq k} \|x_{i+1} - x_i\|_\alpha - \|y - x\|_\alpha \leq \varepsilon$, and*

- *$\{(1 - t)x_i + tx_{i+1} : t \in (0, 1)\} \cap B = \varnothing$ for each $i = 0, \ldots, k$*

*Proof.* Let $v = (x + y)/2$ denote the midpoint between $x$ and $y$, and consider the definable set:

$$P := \{v + w : \langle w, y - x \rangle = 0\} \cap X \subseteq \mathbb{R}^n$$

So $P$ is the intersection of $X$ with the affine space passing through the midpoint $v$ and orthogonal to the segment $y - x$; it follows $\dim P = n - 1$ since $X$ is open. We want to show that there are enough points of $P$ which are arbitrarily close to $v$ such that the pair of line segments $[x, z]$ and $[z, y]$ each have finite intersection with $B$. For this, we can consider the "bad" points of $P$:

$$P_1 := \{z \in P : \dim(([x, z] \cup [z, y]) \cap B) = 1\}$$

Then $P_1$ is a definable set by *Definability of dimension* (Van den Dries, 1998, 4.1.5); it suffices to show that $\dim P_1 < n - 1$. For this, first trim the set $B$ down:

$$B' := B \cap \bigcup_{z \in P_1} [x, z] \cup [z, y]$$

and consider the definable surjection:

$$f \;:\; B' \to P_1, \quad b \mapsto f(b) \;:=\; \text{the unique } z \in P_1 \text{ such that } b \in [x,z] \cup [z,y]$$

Note that by construction we have:

$$P_1 \;=\; \{z \in P_1 : \dim f^{-1}(z) = 1\}$$

and thus by the *Dimension Formula* (Van den Dries, 1998, 4.1.6(ii)) we have:

$$1 + \dim P_1 \;=\; \dim f^{-1}[P_1] \;=\; \dim B' \;<\; n$$

and thus $\dim P_1 < n - 1$. $\qquad\qquad\square$

Note: the lemma is still true without assuming that $X$ is definable.

A SPECIAL CASE OF THE MAIN THEOREM

**Lemma 33.** *If $f : X \to \mathbb{R}^m$ is definable, then:*

$$L^{(\alpha,\beta)}(f,X) \;=\; \sup_{x \in \mathrm{Diff}(f)} \|J_f(x)\|_{\alpha,\beta}$$

*Proof.* Set $L_0 := L^{(\alpha,\beta)}(f,X)$ and $L_1 := \sup_{x \in \mathrm{Diff}(f)} \|J_f(x)\|_{\alpha,\beta}$. The inequality $L_0 \geq L_1$ is clear by Lemma 22. To show $L_0 \leq L_1$, we may assume that $L_1$ is finite. Let $x, y \in X$ be arbitrary distinct points and let $\varepsilon > 0$ be arbitrary; it suffices to show:

$$\frac{\|f(y) - f(x)\|_\beta}{\|y - x\|_\alpha} \;\leq\; L_1 + \varepsilon$$

Set $\varepsilon' := \varepsilon \|y - x\|_\alpha / L_1$ and apply the Path Lemma 32 to obtain $k \geq 0$, points $x_0 = x, x_1, \ldots, x_k, x_{k+1} = y$ in $X$ such that:

- $\sum_{0 \leq i \leq k} \|x_{i+1} - x_i\|_\alpha - \|y - x\|_\alpha < \varepsilon'$, and

- $\{(1-t)x_i + t x_{i+1} : t \in (0,1)\} \cap B = \varnothing$ for each $i = 0, \ldots, k$, where $B := X \setminus \mathrm{Diff}(f)$; note that $\dim B < n$.

Then we have:

$$\|f(y) - f(x)\|_\beta \;\leq\; \sum_{0 \leq i \leq k} \|f(x_{i+1}) - f(x_i)\|_\beta \quad \text{by Triangle inequality}$$

$$\leq\; \sum_{0 \leq i \leq k} L_1 \|x_{i+1} - x_i\|_\alpha \quad \text{by Lemma 25}$$

$$<\; L_1 \|y - x\|_\alpha + L_1 \varepsilon' \quad \text{by choice of } x_i\text{'s}$$

Thus:

$$\frac{\|f(y) - f(x)\|_\beta}{\|y - x\|_\alpha} \;<\; L_1 + \frac{L_1 \varepsilon'}{\|y - x\|_\alpha} \;=\; L_1 + \varepsilon \qquad\qquad\square$$

As an application, can now prove Theorem 3 in the case that $f$ is definable:

**Corollary 34.** *If $f$ is definable, then:*

$$L^{(\alpha,\beta)}(f,X) \;=\; \sup_{x \in X, G \in J_f^c(x)} \|G\|_{\alpha,\beta}$$

*Proof.* Immediately follows from Lemmas 33 and 12. $\qquad\qquad\square$

## GOOD MAPS AND THE EQUIVALENCE LEMMA

In this subsection, to simplify notation, given a set-valued map $D : X \rightrightarrows \mathbb{R}^{m \times n}$, we set:

$$s(D) := \sup_{x \in X, G \in D(x)} \|G\|_{\alpha, \beta}$$

**Definition 35.** *We say a set-valued map $D : X \rightrightarrows \mathbb{R}^{m \times n}$ is **good** for $f$ if:*

- $D(x) \subseteq J_f^c(x)$ *for every $x \in X$, and*

- $D(x)$ *is nonempty for almost every $x \in X$*

**Example 36.** *Here are the two main examples of good maps for $f$:*

- *the full Clarke Jacobian $J_f^c : X \rightrightarrows \mathbb{R}^{m \times n}$ is good for $f$*

- *the usual Jacobian:*

$$J_f : X \rightrightarrows \mathbb{R}^{m \times n}, \quad x \mapsto J_f(x) := \begin{cases} \{J_f(x)\} & \textit{if } x \in \mathrm{Diff}(f) \\ \varnothing & \textit{otherwise} \end{cases}$$

  *is also good for $f$, by Rademacher's Theorem 29*

If $D_0, D_1$ are good maps for $f$, then we define:

$$D_0 \leq D_1 \quad :\Longleftrightarrow \quad D_0(x) \subseteq D_1(x) \quad \text{for every } x \in X$$

The binary relation $\leq$ endows the collection of all good maps for $f$ with the structure of a *partial order*; in fact, this partial order is always a *join-semilattice* (with join operation given by the union $D_0 \vee D_1 := D_0 \cup D_1$). The following is obvious:

**Lemma 37.** *If $D_0 \leq D_1$ are good maps for $f$, then $s(D_0) \leq s(D_1)$.*

Under the assumption of definability, good maps have the following property:

**Lemma 38.** *If $f$ is definable and $D$ is good for $f$, then $D(x) = \{J_f(x)\}$ for almost every $x \in X$.*

*Proof.* Since $f$ is $C^1$ at $x$ for almost every $x \in X$ by Lemma 24, it follows that $J_f^c(x) = \{J_f(x)\}$ for almost every $x \in X$ by Lemma 28, and thus $D(x) = \{J_f(x)\}$ for almost every $x \in X$ by the definition of *good map*. $\square$

It follows immediately that when $f$ is definable, then the partial order $\leq$ is a lattice:

**Lemma 39.** *If $f$ is definable and $D_0, D_1$ are good maps for $f$, then $D := D_0 \cap D_1$ is a good map for $f$.*

**Lemma 40** (Equivalence lemma)**.** *If $f$ is definable and $D_0, D_1$ are two set-valued maps which are good for $f$, then $s(D_0) = s(D_1)$.*

*Proof.* It suffices to show that $s(D) = s(J_f^c)$, where $D$ is an arbitrary good map for $f$. Define $D' := D \cap \{J_f(x)\}$, which is also good map for $f$ by Lemma 39 and Example 36; it suffices to show $s(D') = s(J_f^c)$, by Lemma 37. To set notation, consider:

- $S := \mathrm{domain}(D') = \{x \in X : D'(x) \neq \varnothing\}$, so $S \subseteq \mathrm{Diff}(f)$

- $N := X \setminus S$, so $N$ has Lebesgue measure zero

- $J := J_f|_S$; thus $J : S \to \mathbb{R}^{m \times n}$ fits the setup of Lemma 12

- thus we may further define $D_J : X \rightrightarrows \mathbb{R}^{m \times n}$ to be $D_J(x) := \{\lim_{i \to \infty} J(x_i) : x_i \to x, x_i \in S\}$

Then:

$$
\begin{aligned}
s(D') &= \sup_{x \in S} \|J(x)\|_{\alpha,\beta} \quad \text{by definition} \\
&= \sup_{x \in X, G \in \mathrm{conv}(D_J(x))} \|G\|_{\alpha,\beta} \quad \text{by Lemma 12} \\
&= s(J_f^{c,N}) \quad \text{by definition} \\
&= s(J_f^c) \quad \text{by Lemma 31} \hspace{4cm} \square
\end{aligned}
$$

THE MAIN THEOREM

We may now prove the main Theorem 6:

**Theorem 41.** *If $f$ is definable and $D$ is good for $f$, then:*

$$
L^{(\alpha,\beta)}(f, X) = \sup_{x \in X, G \in D(x)} \|G\|_{\alpha,\beta}
$$

*Proof.* By the Equivalence Lemma 40 it suffices to prove the stated equality for at least one $D$ which is good for $f$. Let $D$ be the good map $\{J_f(x)\}$ (cf. Example 36); then the equality holds for this $D$ by Lemma 33. $\square$

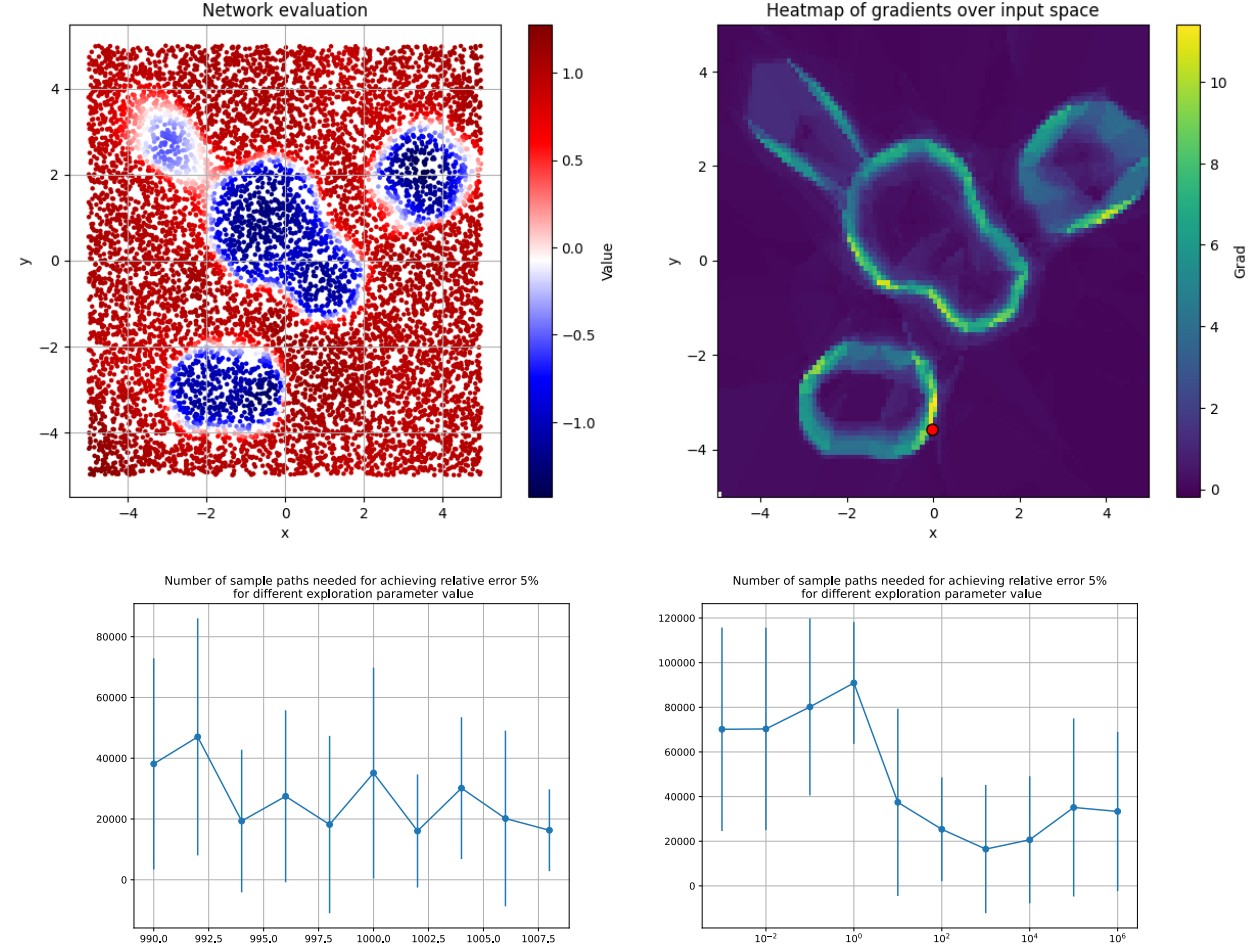

Figure 6: Sensitivity to parameter $c$ of the upper-confidence bound equation 3.3 on an example from the running example (cf. Figure 1). Top: the neural network. Bottom left: sensitivity to small perturbations of the optimal parameter $c$. Bottom right: number of samples required to reach a 5% relative error as a function of the parameter $c$.

## B  SENSITIVITY AND ROBUSTNESS

To study the sensitivity and robustness of the method, we have developed additional experiments in `https://anonymous.4open.science/r/submission-full-code-3C31`. The experiments were run on a depth-7 neural net with hidden layers 32-wide, trained on the same toy dataset as in the running example (cf. Figure 1).

In particular, we focus on hyperparameter $c$ of the upper-confidence bound equation 3.3. From Figure 6, it becomes apparent that the optimal value for $c$ in this particular case is quite large, circa 1000. This is perhaps to be expected: it is clear that the model was trained rather hard, possibly even overfitted, which makes the gradient space (plotted on the top right) quite smooth, requiring more exploration. Having said that, it opens up the question of the robustness of the algorithm with respect to the choice of $c$.

First, let us mention that the consistency of the estimator suggests that for any value of $c$, and for any relative-error threshold, some number of samples will suffice to estimate the modulus of local continuity within that relative-error threshold. For values of $c$ under 10.0, the number of samples required to reach 5% relative-error threshold, sometimes

exceeded 100k samples. For values of $c$ in the range from 10 to 1,000,000, the 5% relative-error threshold has reliably been reached within 100k samples.

Second, let us comment on the sensitivity of the number of samples to the parameter $c$. From the bottom-left subplot in Figure 6, which utilizes a linear scale on the horizontal axis, it is clear that the performance is not particularly sensitive to the perturbation of the exploration parameter $c$. From the bottom-right subplot in Figure 6, which utilizes a logarithmic scale on the horizontal axis, it is clear that underestimating the optimal value of the exploration parameter $c$ by 3 orders of magnitude can increase the number of samples required to achieve a given relative error substantially. On the other hand, overestimating the optimal value of $c$ even by 3 orders of magnitude is not resulting in major performance degradation, since Algorithm 3 overexplores the input spaces, similar to Algorithm 1, whose performance is respectable.

