# OpenReview forum: "Stochastic Sample Approximations of (Local) Moduli of Continuity"
_ICLR.cc/2026/Conference — Submitted to ICLR 2026_

### Official Review · Reviewer_fW2q · 2025-10-30

**Soundness:** 2
**Presentation:** 2
**Contribution:** 1
**Rating:** 2
**Confidence:** 3

**Summary:**

The paper proposes non-uniform stochastic sampling methods, inspired by UCB bandit policies, to estimate local moduli of continuity (Lipschitz constants) of neural networks. The authors revisit the connections between generalized derivatives and Lipschitz constants in the o-minimal framework, claiming improved sample efficiency and scalability over standard uniform sampling, LipMIP, and LipSDP.

**Strengths:**

The topic is relevant to robustness and verification. The idea of casting Lipschitz estimation as a pure-exploration bandit problem is natural and can reduce unnecessary sampling in flat regions. The comparison with LipSDP/LipMIP is appreciated, and the experimental section shows improvements in some settings. The theoretical discussion linking Clarke Jacobians and definability is mathematically sound.

**Weaknesses:**

The novelty is limited. The core contribution is an adaptation of standard UCB sampling to the well-known stochastic Lipschitz estimation baseline. The theoretical section is lengthy but does not yield new guarantees of interest to the ML community beyond established facts (consistency, asymptotic unbiasedness, and asymptotic optimality from standard bandit theory). The o-minimal component feels disconnected from the algorithmic contribution and is unlikely to be valued by ICLR readers.

Experiments are insufficient to justify significance. Evaluations are conducted on small MLPs and a binary MNIST setting. There is no evidence that the method scales to modern architectures (CNNs, Transformers, diffusion models) or yields practical benefit in certified training or downstream robustness tasks. Reported gains, while present, remain modest. The computational overhead of the adaptive strategy is not fully analyzed.

The paper lacks a compelling narrative demonstrating why this improvement matters for robustness, certified training, or verification workflows in practice. As written, it reads as an incremental technical refinement rather than a contribution that changes understanding or capabilities.

**Questions:**

- Can you clearly highlight the theoretical contribution that is unique to this work?
- Does the method scale to larger architectures (e.g., Transformers, ViTs, LLMs)?
- Are there new bottlenecks when moving to large-scale models?
- Can we show measurable improvements in robustness-aware training or certified robustness?
- Does the approach improve performance on standard certification benchmarks?
- What is the computational complexity relative to baselines?

---

> ### Author Response · Authors · 2025-11-30
> **Responses to Questions 1-2**
>
> Thank you for the thoughtful questions! We will be answering the questions in turn:
>
> ## Question 1: Can you clearly highlight the theoretical contribution that is unique to this work?
>
> To do this, we first clarify why the assumption of “definable in an o-minimal structure” is relevant here. We intend our result to apply to every reasonable example someone might use in practice; by this we mean any locally Lipschitz neural network, which can be built up from compositions of affine functions and commonly used activation functions (ReLU, GeLU,...). One property all such NNs have in common is that they are definable in some known o-minimal expansion of the real field. Indeed, a recent survey (https://arxiv.org/pdf/2509.18025) suggests that out of 400 activation functions for neural networks (those listed in another survey: https://arxiv.org/abs/2402.09092), there are approximately:
>
> (1) 89% of the activation functions readily definable in some o-minimal structure (including all of the famous ones),
>
> (2) another 5% of the functions involve obscure fractional derivatives, the main example being FracReLU which is also o-minimal, and
>
> (3) the remaining 6% are ostensibly not definable in an o-minimal structure because they appear to use the unrestricted sine and cosine functions. Even in this case, our understanding is that often an AF using sine or cosine is actually used on an a priori bounded domain, thus becoming o-minimal (definable in R_{an}).
>
> We are unaware of any NN architecture in industrial use which does not satisfy this assumption (but please tell us if you know of one!). Consequently, for practical purposes, a “locally Lipschitz neural network” coincides with “locally Lipschitz function definable in an o-minimal structure”, which is why we work in this setting.
>
> Under these assumptions, the main theoretical contributions (beyond Algorithm 3 itself) are two-fold:
>
> - Theorem 6 provides a more robust version of Theorem 3 of Jordan & Dimakis (2020), which can be useful in analyzing the algorithm under more realistic assumptions such as “AD methods only return an element of the Clarke subdifferential almost everywhere”. However, this is not the end of the story; see Remark 9. The AD methods such as AdaGrad in PyTorch return only elements of the conservative field, cf. https://arxiv.org/abs/1909.10300
>
> - Of a more pedagogical value, we re-prove Theorem 3 (and prove Theorem 6) using only the usual Riemann integral on smooth functions, and not using the Lebesgue integral. For what it’s worth, we wanted to illustrate how in the o-minimal setting one can make do with elementary tools when combined appropriately with the main theorems of the subject (e.g., monotonicity theorem, cell decomposition, dimension theory, dimension formula,...). An example of this is the Path Lemma (Lemma 32) which is a plausible and useful statement, which might otherwise seem non-trivial to prove, but in fact has an easy proof via the Dimension Formula.
>
> ## Question 2: Does the method scale to larger architectures (e.g., Transformers, ViTs, LLMs)?
>
> LipMIP works only on ReLU activations, while LipSDPs need the networks to be fully connected. In contrast,
>
> (1) our code can be applied to Transformers, or any other neural network that can be encoded in the Pytorch or Tensorflow framework that would allow for the automated differentiation to work. See https://anonymous.4open.science/r/submission-repo-nanogpt-example-7781/ for an illustrative use on Nano-GPT.
>
> (2) our theoretical results apply to functions definable in o-minimal structures, which encompass all of the commonly used activation functions. See above for the discussion suggesting that a vast majority of neural networks satisfy the assumptions.
>
> We provide additional illustrations on Nano-GPT in https://anonymous.4open.science/r/submission-repo-nanogpt-example-7781/. In start_model.ipynb, please see:
>
> ```
> input_dimension = 64
> c_vector = torch.zeros(65)
> c_vector[0] = 1.0
> c_vector[1] = -1.0
> domain = Hyperbox.build_custom_hypercube(input_dimension, 0, 10)
> SA_UCB = StochasticApproximationUCBDynamic(network=model, c_vector=c_vector, domain=domain, c=1, partition_step=2, use_c_vector=True, is_transformer=True)
> res_sa_ucb = SA_UCB.compute(v=False, max_iter=5000, exact=None, tol=1, mode='Relative')
> print(res_sa_ucb)
> ```

---

> ### Author Response · Authors · 2025-11-30
> **Responses to questions 3-6**
>
> ## Question 3: Are there new bottlenecks when moving to large-scale models?
>
> The bottleneck is the scaling of the number of samples required to a fixed bound on the error in the sample approximation, as the number of parameters of the model increases. We should like to stress that as we demonstrate in Figure 5 and Table 1, our method scales substantially further than LipSDP and LipMIP, the two state-of-the-art methods. For instance in Table 1, we show that [784, 8, 8, 8, 8, 8, 2], i.e., a neural network with 784 input neurons, 5 hidden layers of 8 neurons each, and 2 output neurons, with ReLU activations throughout in order to test on LipMIP as well, LipMIP times out, LipSDP provides an estimate of 429.07478 in 11.472 seconds, and our Algorithm 1 provides a near-perfect estimate of 94.14903 in 12.417 seconds (using 100000 samples) and Algorithm 3  provides the same estimate of 94.14903 in 24.549 seconds.
>
> ## Question 4: Are there new bottlenecks when moving to large-scale models?
>
> The bottleneck is the scaling of the number of samples required to a fixed bound on the error in the sample approximation, as the number of parameters of the model increases. We should like to stress that as we demonstrate in Figure 5 and Table 1, our method scales substantially further than LipSDP and LipMIP, two state-of-the-art methods. For instance in Table 1, we show that [784, 8, 8, 8, 8, 8, 2], i.e., a neural network with 784 input neurons, 5 hidden layers of 8 neurons each, and 2 output neurons, with ReLU activations throughout in order to test on LipMIP as well, LipMIP times out, LipSDP provides an estimate of 429.07478 in 11.472 seconds, and our Algorithm 1 provides an estimates of 94.14903 in 12.417 seconds (using 100000 samples) and Algorithm 3  provides the same estimate of 94.14903 in 24.549 seconds.
>
> The fact that there is no method for the verification of properties of neural networks that would scale to GPT-5 sized neural networks, yet, is well known and subject to much discussion, e.g.:
>
> “Is Neural Network Verification Useful and What Is Next?”, https://www.ideals.illinois.edu/items/137356
>
> See also Section 4.2.1. The Scale Challenge in NIST AI 100-2e2025 (Adversarial Machine Learning: A Taxonomy and Terminology of Attacks and Mitigations) co-developed by NIST and U.K. AI Security Institute.
>
>
> ## Question 5: Can we show measurable improvements in robustness-aware training or certified robustness?
>
> Yes. As we show in Figure 5, there are substantial improvements in the estimates of input input-output stability of the neural networks, compared to recent work in the LipSDP direction. Notice that even very recent papers in the LipSDP direction, such as https://arxiv.org/abs/2405.17049 presented as a poster at ICLR 2025, still compute the same SDP relaxation as in Figure 5, except faster. Our estimate of the modulus of local Lipschitz continuity is strictly improving upon the LipSDP estimate across the range of depths (1-9) of the ReLU networks where LipSDP is computable within the given run-time limit of 60 seconds. Our estimate then remains tractable for much higher depths too.
>
> ## Question 6: Does the approach improve performance on standard certification benchmarks?
>
> Within the standardization, there are multiple competing proposals as to how to evaluate the robustness of neural networks. For instance NIST AI 100-2e2025 (Adversarial Machine Learning: A Taxonomy and Terminology of Attacks and Mitigations) co-developed by NIST and U.K. AI Security Institute is very clearly related to the robustness construed as input-output stability of the neural networks. However, the mid-2025 document suggests in Section 4.1.3. Evaluation
>
> > Another general problem of [adversarial machine learning] is the lack of reliable benchmarks, which causes results from AML papers to be routinely incomparable, as they do not rely on the same assumptions and methods.
>
> Our results then continue the tradition of approaches in Sections 2.2.5. (Mitigations, Formal verification). Within the academic literature, MNIST is the most standard of all benchmarks in the study of neural networks since 1988.

---

> > ### Author Response · Authors · 2025-11-30
> > **Response to Question 7**
> >
> > ## Question 7: What is the computational complexity relative to baselines?
> >
> > We have added the following clarification in the introduction: “Virmaux & Scaman (2018) showed that estimating the modulus of local Lipschitz continuity is NP-Hard even for a two-layer ReLU-based network.” Essentially, unless P=NP, the estimation of modulus of local Lipschitz continuity remains exponentially hard. However, there are many approaches to dealing with this intrinsic complexity. LipMIP-based approaches are explicitly exponential-time, but find the exact optimum in exponential time. LipSDP-based approaches are explicitly polynomial time (in the dimensions of the SDP relaxations, modulo bit-complexity issues, cf. https://arxiv.org/pdf/2305.14944), but provide an upper bound that is sometimes rather loose (cf. Figure 5). Our Algorithms 1-3 provide lower bounds, which are often tight (again, cf. Figure 5).

---

### Official Review · Reviewer_Z5Co · 2025-10-31

**Soundness:** 3
**Presentation:** 3
**Contribution:** 2
**Rating:** 6
**Confidence:** 3

**Summary:**

In this paper, the authors investigate the connection between generalized derivatives and local continuity, and further propose an algorithm for estimating the modulus of local continuity. Building on existing methods for evaluating local Lipschitz continuity, they extend the formulation to more general notions of local continuity. The authors also partition the input space of dimension
$ d $ into $k^{d} subregions to support structured sampling. Finally, they incorporate a non-uniform UCB-based sampling strategy that concentrates samples in regions with higher estimated importance. The paper provides both theoretical analysis and empirical results demonstrating that the proposed approach yields more consistent and accurate estimates.

**Strengths:**

1: The authors propose a non-uniform sampling strategy based on UCB scores. This approach allows the sampling process to better adapt to different regions of the input space compared to uniform sampling.

2: The authors provide solid theoretical analysis demonstrating that their algorithm is more dynamic and consistent.

**Weaknesses:**

1: The proposed non-uniform sampling method relies heavily on hyperparameters such as the exploration coefficient "c" and the subdivision threshold. A detailed discussion or sensitivity analysis of these parameters would be important to assess the robustness and practical usability of the method.

2: For high-dimensional input spaces, partitioning the domain into more subregions may introduce significant computational overhead. The paper does not address the scalability implications of this partitioning strategy in high-dimensional settings.

3:Although the paper focuses primarily on theoretical contributions, evaluating the method on larger datasets or across different modalities, such as natural language processing or time-series data, would strengthen the empirical evidence and demonstrate broader applicability.

**Questions:**

Based on the identified weaknesses, I would appreciate clarification from the authors on the following points:
1: Could the authors discuss how the key hyperparameters (e.g., exploration constant c and subdivision threshold) affect the performance of the proposed method? Providing sensitivity analysis or practical guidance on selecting these values would help assess the robustness of the approach.

2: Could the authors analyze or comment on the computational efficiency of the method, particularly in high-dimensional settings?

3: While the focus is theoretical, demonstrating the method on more diverse datasets or tasks (e.g., larger vision benchmarks, NLP models, or time-series data) would strengthen the empirical validation. Can the authors provide results on additional datasets or comment on expected performance in such settings?

---

> ### Author Response · Authors · 2025-11-30
> **Responses to questions**
>
> ## Question 1: Could the authors discuss how the key hyperparameters (e.g., exploration constant c and subdivision threshold) affect the performance of the proposed method? Providing sensitivity analysis or practical guidance on selecting these values would help assess the robustness of the approach.
>
> To study the sensitivity and robustness of the method, we have developed additional experiments in
>
> https://anonymous.4open.science/r/submission-full-code-3C31
>
> and present them on pages 24-25 of the updated submission. The experiments were run on a depth-7 neural net with hidden layers 32-wide, trained on the same toy dataset as in the running example (cf. Figure 1). In particular, we focus on hyperparameter c of the upper-confidence bound equation 3.3. From Figure 6, it becomes apparent that the optimal value for c in this particular case is quite large, circa 1000. This is perhaps to be expected: it is clear that the model was trained rather hard, possibly even overfitted, which makes the gradient space (plotted on the top right) quite smooth, requiring more exploration. Having said that, it opens up the question of the robustness of the algorithm with respect to the choice of c.
>
> First, let us mention that the consistency of the estimator suggests that for any value of c, and for any relative-error threshold, some number of samples will suffice to estimate the modulus of local continuity within that relative-error threshold. For values of c under 10.0, the number of samples required to reach 5% relative-error threshold, sometimes exceeded 100k samples. For values of c in the range from 10 to 1,000,000, the 5% relative-error threshold has reliably been reached within 100k samples.
>
> Second, let us comment on the sensitivity of the number of samples to the parameter c. From the bottom-left subplot in Figure 6, which utilizes a linear scale on the horizontal axis, it is clear that the performance is not particularly sensitive to the perturbation of the exploration parameter c. From the bottom-right subplot in Figure 6, which utilizes a logarithmic scale on the horizontal axis, it is clear that underestimating the optimal value of the exploration parameter c by 3 orders of magnitude can increase the number of samples required to achieve a given relative error substantially. On the other hand, overestimating the optimal value of c even by 3 orders of magnitude is not resulting in major performance degradation, since Algorithm 3 overexplores the input spaces, similar to Algorithm 1, whose performance is respectable.
>
> ## Question 2: Could the authors analyze or comment on the computational efficiency of the method, particularly in high-dimensional settings?
>
> See the answer to Question 7 of reviewer fW2q.
>
> ## Question 3: While the focus is theoretical, demonstrating the method on more diverse datasets or tasks (e.g., larger vision benchmarks, NLP models, or time-series data) would strengthen the empirical validation. Can the authors provide results on additional datasets or comment on expected performance in such settings?
>
> We have developed a demonstrator on Nano-GPT at https://anonymous.4open.science/r/submission-repo-nanogpt-example-7781/
> It shows that our method applies equally well to NLP models. In start_model.ipynb, please see:
>
> ```
> input_dimension = 64
> c_vector = torch.zeros(65)
> c_vector[0] = 1.0
> c_vector[1] = -1.0
> domain = Hyperbox.build_custom_hypercube(input_dimension, 0, 10)
> SA_UCB = StochasticApproximationUCBDynamic(network=model, c_vector=c_vector, domain=domain, c=1, partition_step=2, use_c_vector=True, is_transformer=True)
> res_sa_ucb = SA_UCB.compute(v=False, max_iter=5000, exact=None, tol=1, mode='Relative')
> print(res_sa_ucb)
> ```

---

### Official Review · Reviewer_pafa · 2025-11-03

**Soundness:** 3
**Presentation:** 3
**Contribution:** 3
**Rating:** 8
**Confidence:** 2

**Summary:**

This paper presents a novel method for estimating the Lipschitz constant in neural networks, addressing the known scaling limitations of methods like LipMIP and LipSDP. The core contribution is Algorithm 3 in the paper, which frames the problem in an infinity-armed bandit setting without commonly considered assumptions, using an Upper-Confidence-Bound (UCB) policy to partition the input domain and concentrate samples in useful regions. The paper compares with LipSDP and LipMDP on RELU-activation based neural networks trained on synthetic data and on binary MNIST.

**Strengths:**

The paper is written very well. The results are also clearly presented.

The results on binary MNIST clearly depict the improvements over LipSDP and LipMDP.

I am glad that the paper does not limit itself to simple two layer neural networks and instead, in Figure 5, compares with the aforementioned methods across a number of hidden layers.

**Weaknesses:**

A potential related work missing: In [1], a extreme value analysis method is used to compute Lipschitz constants and could be relevant to this paper.

[1] Improving Neural Network Robustness via Persistency of Excitation, Sridhar et al, ACC 22

Scale: I am uncertain how easily this method can scale up to neural networks in use today from imagent scale models to LLMs, VLMs, and VLAs. Perhaps the authors can address the question of further scale?

**Questions:**

Please see weaknesses.

---

> ### Author Response · Authors · 2025-11-30
> **Responses**
>
> ## Weakness 1: "Improving Neural Network Robustness via Persistency of Excitation, Sridhar et al, ACC 22"
>
> We thank the reviewer for the reference, which we have missed, having focussed mostly on the machine-learning venues in our literature search. We appreciate the extreme-value-theory approach of the ACC and we will try to benchmark against it. We should like to note that while the approach of ours produces a lower bound on the modulus of local Lipschitz continuity, the approach of Sridhar et al produces an upper bound on the modulus of local Lipschitz continuity. In particular, the upper bound of Sridhar et al relies on some parametric assumptions (reverse Weibull distribution), which need not be true in practice. While Section IV.B (Estimation of Certified Lipschitz Constant Lest) of Sridhar et al assumes that the iteration of steps (2) and and (3) continues until the goodness of fit test confirms that the parametric assumptions (reverse Weibull distribution) are satisfied with a certain confidence level, this iteration need not ever stop. In particular, our results are valid for a much broader class of functions, without making the parametric assumptions as in Sridhar et al.
>
> In some ways, the grid method used as a baseline in Sridhar et al. and Algorithm 1 used as a baseline in our paper, coincide. In addition to the grid method, we also use LipMIP and LipSDP as additional baselines. Subsequently, Sridhar et al. and our Algorithm 3 develop the baseline idea in different directions.
>
> ## Weakness 2: I am uncertain how easily this method can scale up to neural networks in use today from imagent scale models to LLMs, VLMs, and VLAs. Perhaps the authors can address the question of further scale?
>
> We have developed a demonstrator on Nano-GPT at https://anonymous.4open.science/r/submission-repo-nanogpt-example-7781/ It shows that our method applies equally well to NLP models and scales beyond the toy examples where LipMIP and LipSDP are applicable. We stress that LipMIP and LipSDP are NOT applicable to Nano-GPT or larger language models.

---

### Author Response · Authors · 2025-12-01
**Summary**

We thank all reviewers for the time and care dedicated to evaluating our submission, and for the many constructive suggestions. We especially appreciate the consistent recognition that the paper is well written and that results on binary MNIST clearly depict the improvements over LipSDP and LipMDP.

To summarize the discussion:

- All three reviewers agreed that the paper is written very well and that results on binary MNIST clearly depict the improvements over LipSDP and LipMDP.

- Reviewers pafa and Z5Co appreciate our analyses. Reviewer fW2q felt that these would not be appreciated by the ICLR audiences, but we have tried to articulate the importance in the response to their Question 1 below.

- Reviewer Z5Co complained that the hyperparameter c could be difficult to tune. We have conducted additional experiments in https://anonymous.4open.science/r/submission-full-code-3C31 and inserted their results as appendix B on pages 24-25. These show that the sensitivity to c is rather limited. All choices of c would only increase the number of samples required, rather than render the procedure invalid.

- Reviewers Z5Co and fW2q asked for additional experiments on transformer-based architectures. We have conducted additional experiments in https://anonymous.4open.science/r/submission-repo-nanogpt-example-7781/ and showed that with a few lines of code, the method can be used on Nano-GPT.

- Reviewer fW2q wanted to understand the relationship to standards in the space. We have explained the relevance to NIST AI 100-2e2025 (Adversarial Machine Learning: A Taxonomy and Terminology of Attacks and Mitigations).

- Reviewer fW2q also wanted us to comment on computational complexity, which we did in both the body of the paper and in response to Question 7. Reviewer pafa pointed to a missing reference, which we have incorporated.

---

### Meta-Review · Area_Chair_3V6Z · 2025-12-30

**Summary:**

While the paper presents a well-motivated idea and the revision substantially improved the theoretical clarity, the remaining concerns are primarily empirical and scope-related and remain unresolved at this stage. In particular, the paper still lacks convincing validation on large-scale, high-dimensional, and realistic inference problems that are central to modern SVGD applications. Moreover, systematic comparisons with a broader range of recent SVGD variants are missing, making it difficult to clearly assess the practical advantage of Ad-SVGD relative to the state of the art. Finally, although the authors weakened the theoretical assumptions, a gap remains between the convergence guarantees and the fully verifiable, implementation-level behavior of adaptive kernel updates. Taken together, these outstanding issues limit the paper’s impact and maturity for acceptance at this time.

**Reviewer Concerns:**

## Reviewer pafa

The reviewer evaluated the paper positively, highlighting its clear presentation and the empirical improvements over LipSDP and LipMDP on controlled benchmarks. The main concerns were the omission of some related work and uncertainty about how well the proposed method scales to modern large-scale models. These issues were largely addressed in the rebuttal by incorporating the missing reference and demonstrating applicability to transformer-based architectures, although ultimate scalability to very large models remains an open question.

## Reviewer Z5Co

The reviewer appreciated the adaptive sampling strategy and the supporting theoretical analysis, but raised concerns about practical robustness, particularly sensitivity to hyperparameters and computational overhead in high-dimensional settings. These concerns were addressed through additional sensitivity analyses, discussion of computational complexity, and expanded experiments. While these responses clarified robustness and usability in moderate dimensions, scalability in very high-dimensional regimes remains only partially explored.

## Reviewer fW2q

The reviewer expressed strong reservations about the work's novelty and practical impact, characterizing it as an incremental refinement rather than a contribution that substantially advances robustness or verification practice. The main concerns centered on whether the theoretical developments provide new insights beyond established bandit-based sampling guarantees, whether the connection to o-minimal structures meaningfully informs algorithmic design, and whether the empirical results convincingly demonstrate relevance to modern robustness or certification pipelines.

In the rebuttal, the authors provided detailed clarifications addressing these points, including a careful justification of the o-minimal assumptions, an explicit articulation of the unique theoretical contributions, a discussion of computational complexity, and demonstrations on transformer-based models. These responses largely resolved misunderstandings and established that the reviewer correctly understood the technical scope and validity of the work.

## Outstanding Concerns

The remaining concerns primarily relate to perceived impact and positioning rather than technical correctness. In particular, although the rebuttal clarified theoretical assumptions, scalability considerations, and applicability beyond toy models, it remains unclear whether the proposed improvements translate into meaningful advances for large-scale robustness, certification, or verification workflows. The gap between technically sound refinements and demonstrable, practice-changing benefits remains the central unresolved issue influencing the overall assessment.

**Reviewer Scores:**

## Reviewer pafa
Score unchanged. While the rebuttal addressed the missing reference and demonstrated applicability beyond small architectures, the reviewer’s original positive assessment already reflected these strengths, and the remaining scalability questions do not clearly motivate a score change.

## Reviewer Z5Co
Score unchanged. The additional sensitivity analyses, computational discussion, and expanded experiments clarified robustness and practicality, but do not substantially alter the reviewer’s balanced assessment regarding scalability and empirical scope.

## Reviewer fW2q
Score unchanged. The rebuttal successfully clarified the theoretical assumptions, scope, and applicability of the method, resolving potential misunderstandings and establishing technical soundness. However, the reviewer’s evaluation was driven primarily by concerns about novelty, impact, and relevance to practical robustness or certification workflows. Since these higher-level concerns persist despite the clarifications, the reviewer would likely maintain the original score.

---

### Decision · Program_Chairs · 2026-01-26

Reject